



# Evaluating a Global Soil Moisture dataset from a Multitask Model (GSM3 v1.0) for current and emerging threats to crops

Jiangtao Liu[1], David Hughes[2,3,4], Farshid Rahmani[1], Kathryn Lawson[1], Chaopeng Shen[1]

[1]Department of Civil and Environmental Engineering, The Pennsylvania State University, University Park, PA, USA
[2]Department of Entomology, The Pennsylvania State University, University Park, PA, USA
[3]Department of Biology, The Pennsylvania State University, University Park, PA, USA
[4]The Current and Emerging Threat to Crop Innovation Lab, The Pennsylvania State University, University Park, PA, USA

*Correspondence to*: Chaopeng Shen (cshen@engr.psu.edu)

**Abstract.** Climate change threatens our ability to grow food for a growing population. There are concurrent droughts and floods happening globally, with the greatest impacts felt in Africa. There is a need for high-quality soil moisture predictions in under-monitored regions like Africa. Yet it is unclear if soil moisture processes are globally similar enough to allow our models to maintain accuracy in unmonitored regions. We present a multitask long short-term memory (LSTM) model that learns simultaneously from global satellite-based and in-situ soil moisture data. This model is evaluated in both random spatial holdout mode and continental holdout mode (trained on some continents, tested on a different one). The model compared favorably to current land surface models, satellite products, and a candidate machine learning model, reaching a global median correlation of 0.792 for the random spatial holdout test. It behaved surprisingly well in Africa and Australia, showing high correlation even when we excluded their sites from the training set, but performed relatively poorly in Alaska where rapid changes are occurring. In all but one continent (Asia), the multitask model in the worst-case scenario test performed better than the soil moisture active passive (SMAP) 9-km product. Factorial analysis shows that the LSTM model's accuracy with representing impacts of terrain aspect, resulting in lower performance for dry and south-facing slopes or wet and north-facing slopes. This knowledge helps us apply the model while understanding its limitations. This model is being integrated into an operational agricultural assistance application which currently provides information to 13 million African farmers.

### 500-character non-technical summary

Under-monitored regions like Africa need high-quality soil moisture predictions to help with food production, but it is not clear if soil moisture processes are similar enough around the world for data-driven models to maintain accuracy. We present a deep-learning-based soil moisture model that learns from both in-situ data and satellite data and performs better than satellite products at the global scale. These results help us apply our model globally while better understanding its limitations.



## 1. Background

Soil moisture is a critical variable that influences a number of natural disasters. As a result, widely available, high-quality soil moisture products can be vital for regions that need aid. Too much soil moisture can prime the landscape for floods (Norbiato et al., 2008), and too little of it for too long can damage or kill crops and native vegetation (Narasimhan and Srinivasan, 2005; Sheffield and Wood, 2008). Moreover, many insect pests lay eggs in soils with certain soil moisture conditions - for example, locusts prefer to lay their eggs in sandy, wet soils

(Hunter-Jones, 1964). In the year 2020, disastrous locust swarms terrorized large swaths of Eastern Africa and Southeast Asia (Baraniuk, 2020; UN WFP, 2020). The knowledge of soil moisture levels can be critical in planning pest control activities, as the immature stages are the best targets for effective control (Ellenburg et al., 2021; Nuwer, 2021). Besides insect pests, pathogenic fungi and bacteria can be heavily influenced by soil moisture, resulting in crop losses. In all of these cases, soil moisture products can be highly valuable in reducing

both current and emerging threats to crops. Finally, the current global crisis in fertilizer availability following the ongoing war in Europe (Bentley et al., 2022) necessitates strategies that increase the efficient use of fertilizer, for which a precise understanding of soil moisture is critical since water availability in the soil affects both plant uptake of fertilizer and fertilizer loss.

Soil moisture is monitored globally by a number of satellite missions as well as simulated globally by multiple land surface hydrologic models, but these products have their limitations. Satellite missions like Soil Moisture Active Passive (SMAP) (Entekhabi, 2010) and Soil Moisture and Ocean Salinity (SMOS) (Kerr et al., 2010), have limited spatial resolution and accuracy. When evaluated in comparison to in-situ data, especially on sparsely-instrumented sites that are outside of the missions' core calibration/validation sites, their error can be high (Al-

Yaari et al., 2017) (also demonstrated later in this work). Land surface models can also produce decent simulations with seamless spatiotemporal coverage (Albergel et al., 2018; Beaudoing et al., 2019; Yang et al., 2011), but they may not be fully exploiting available information, as evidenced by the better performance produced by machine learning models where data are available (Liu et al., 2022; O and Orth, 2021). Both satellite and model products may also have a large bias compared to in-situ data.


Recently, we developed a multiscale time-series deep learning (DL) model that learns simultaneously from satellite and in-situ data and can substantially outperform satellite-based products, a model trained on in-situ data alone, and traditional land surface model simulations (Liu et al., 2022). In a spatial cross-validation test (trained on some sites and tested on others), the multiscale DL model obtained a median correlation (R) of 0.901 when

evaluated by the sparse soil moisture network over the conterminous United States (CONUS), comparing favorably to the SMAP 9km product's R value of 0.762 and Noah model's 0.761, and it had minor bias. This work suggested that many previous simulations have not fully leveraged the available information. In addition, it demonstrated that multiple sources of datasets may each constrain certain aspects of a network and train models that outperform each one of its supervising datasets, i.e., *learning from two teachers can be better than one*. This

multiscale approach can overcome the limitations with each single dataset.

However, it is uncertain if the robust model performance from deep networks in the data-dense CONUS can generalize well to other regions in the world where it is of interest due to potential natural disasters. Typically, the



performance of all kinds of models declines somewhat when applied to neighboring untrained sites (as in a random
holdout test), and then declines substantially when applied in a large region without training data (Feng et al.,
2021; Gauch et al., 2020; Hrachowitz et al., 2013). Sequence-to-sequence deep networks like long short-term
memory (LSTM) (Hochreiter and Schmidhuber, 1997) can give us high predictive performance in a range of
hydrologic tasks (Fang et al., 2017, 2019; Feng et al., 2020; Kratzert et al., 2019; Meyal et al., 2020; Rahmani et
al., 2021b; Shen, 2018; Zhi et al., 2021) because they do not have rigid model structures and can absorb
information more exhaustively from big data. Their functional behaviors are completely shaped by data, and thus
they can be exempt from many errors in previous models' assumptions. On the flip side, in data-sparse regions,
there is a chance that such a strength could become a weakness. In Africa, especially, there are very few in-situ
sites to constrain a model. Recent work has trained LSTM-based global soil moisture models completely on in-
situ sites, for example, the SoMo.ml model (O and Orth, 2021), but this only learns from in-situ locations (O and
Orth, 2021; Science Data Curation Team, 2021). It is not clear if optimality has been reached by such models, or
if a multitask model learning from both satellite and in-situ data could provide further advantages.

Regarding the potential for data-driven models in data-scarce regions, there can be two competing hypotheses.
The optimistic hypothesis is that surface soil moisture dynamics is relatively simple to grasp (compared to the
streamflow prediction problem), quite uniform around the world, and well described by available surface
characterization datasets (soil texture) --- as a result, the hundreds of publicly available sites can thoroughly train
a DL model that generalizes well in space. The more pessimistic hypothesis is that the quality of available inputs,
e.g. soil texture, is low so that the number of sites in the world is far from being sufficient to train a global-scale
DL soil moisture model. Confirming one hypothesis or the other not only influences how we choose a model, but
may also alter our understanding about the complexity of the soil moisture prediction problem.

Given that we would like to have a high-quality product in data-sparse regions like Africa, we ask three research
questions regarding not only the performance of a global-scale LSTM-based soil moisture model, but also the
nature of the soil moisture dynamics:
1. *How well can a LSTM-based soil moisture model perform on the global scale for untrained sites, in comparison
to existing satellite-based and model-based products?*
2. *How well can such a model generalize to highly data-sparse regions, e.g., in an entire continent without data -
- are soil moisture processes homogeneous enough to permit cross-continental model applications*?
3. *What factors control the success or failure of such a model, i.e., can we predict, a priori, if this model can be
successful?*

We developed and trained a multitask LSTM-based model that learns simultaneously from both satellite and in-
situ data. We tested the model in random hold out and cross-continental experiments to learn its strengths and
weaknesses. We then used a stratified analysis to diagnose where the model would likely be successful or
challenged. In the end, we produced a globally-operational surface soil moisture product that can be leveraged by
non-profit organizations at 9-km resolution.



## 2. Data and Methods

### 2.1. The multitask LSTM model.

The multitask model based on the long short-term memory (LSTM) algorithm can be described succinctly as the following:

$$y = LSTM(x, A) \tag{1}$$
$$L = RMSE(y, y^s) + RMSE(y, y^{in}) \tag{2}$$

where $y$ represents simulated soil moisture, $x$ represents dynamic atmospheric forcings, and $A$ represents static landscape attributes. $L$ is the loss function the model tries to minimize, which is based on root-mean-square error (RMSE). $y^s$ represents satellite-based soil moisture products (SMAP L3, 9 km resolution), and $y^{in}$ represents in-
situ data (from the International Soil Moisture Network, ISMN). This model does not use recent observations and is thus suitable for long-term simulations or trend predictions, but could be enhanced for short-term forecasting via data assimilation or data integration (Fang and Shen, 2020; Feng et al., 2020). This multitask loss function means that the simulations will attempt to respect both in-situ data and satellite data. Since LSTM has been described extensively in previous work (Fang et al., 2019; Feng et al., 2020; Liu et al., 2022), we omit its
mathematical descriptions here for brevity. Here, because we are now applying it on a global scale, we chose this multitask scheme over our previous multiscale scheme (Liu et al., 2022) which aggregates many fine-resolution gridcells to match a coarse-resolution gridcell, to reduce computational demand. To avoid overturning the hyperparameters, we inherited most of the parameters from our multi-scale model. Our final parameters were as follows: a mini-batch size of 128, a hidden-state size of 256, a dropout rate of 0.5, an epoch length of 100, and a
sequence length (rho) of 365 days.

### 2.2. The input and training datasets.

We used the SMAP Enhanced L3 Radiometer Global and Polar Grid Daily 9 km EASE-Grid Soil Moisture, Version 5 (SPL3SMP_E) product (O'Neill et al., 2021) as our satellite target, and the International Soil Moisture Network (ISMN) product as our in-situ target (Dorigo et al., 2011, 2013). The input data includes 18 different
meteorological forcings and 17 different static attributes. We obtained leaf area index (LAI), soil temperature, and surface pressure, and others (Table S1 in the Supplementary Material) from the ECMWF Reanalysis v5 (ERA5) (Setchell, 2020). We tried multiple sources of precipitation data, including Multi-Source Weighted-Ensemble Precipitation (MSWEP) (Beck et al., 2019), Global Precipitation Measurement (GPM) (Huffman et al., 2019) and ERA5 precipitation data. Our preliminary results suggested that, in terms of the correlations of the resulting
models, we had this order: MSWEP+GPM+ERA5≈MSWEP>GPM>ERA5. Thus, to allow the model to fully absorb the precipitation information, we include both MSWEP, GPM, and ERA5 in the input data. Albedo data include black sky albedo and white sky albedo from the Moderate Resolution Imaging Spectroradiometer (MODIS) MCD43A3 Version 6 (Schaaf, Crystal and Wang, Zhuosen, 2021). The Land Surface Temperature (LST) dataset includes LST day and LST night data from MODIS Land Surface Temperature/Emissivity Daily
(MYD11A1) Version 6.1 (Wan, Zhengming et al., 2021).

Static terrain attributes included slope, aspect, plane curvature (pcurv), elevation, and roughness from the Global 1,5,10,100-km Topography database (Amatulli et al., 2018). Aspect was determined using the aspect cosine, which is >0 for north-facing and <0 for south-facing slopes in the Northern Hemisphere. We further multiplied



the aspect cosine in the Southern Hemisphere by -1 to reflect the sun's position. Soil physiographic attributes included sand, clay, and silt fractions, and bulk density from the Harmonized World Soil Database v1.2 (HWSD) (FAO et al., 2012; Fischer et al., 2008). Other attributes including land cover (ESA, 2017) and Normalized Difference Vegetation Index (Didan, 2015) were derived from several satellite products. All attributes and their sources are listed in Table S1 in the Supplementary Material.


To train the model and evaluate its performance, we used soil moisture measurements ($m^3m^{-3}$) from the International Soil Moisture Network (ISMN) (Table S2 in Supplementary Material). The ISMN is an international collaboration where soil moisture measurements are collected from dozens of soil moisture networks across the world. We selected site data from ISMN with ~5cm depth and aggregated the hourly data into daily data. We used

a total of 1317 sites, located across Africa (18), Asia (115), Europe (129), the CONUS (969), Alaska (44), and Australia (19). Based on the site clustering in Africa, we divided the data on Africa into North Africa and South Africa according to latitudes 1.8 to 19.3 and -38.9 to -22.0, respectively.

### 2.3. The models and products for comparisons.

We compared the results with the SMAP-L3 enhanced 9-km product (O'Neill et al., 2021), the SMOS-L3 product

(Al Bitar et al., 2017; Support CATDS, 2022), the LPRM_AMSR2_DS_A_SOILM3 product (de Jeu, Richard, 2013; Owe et al., 2008), the NOAH025 (10 cm depth) model from the Global Land Data Assimilation System (GLDAS) (Beaudoing et al., 2019; Rodell et al., 2004), and another machine learning model, SoMo.ml (O and Orth, 2021). SMAP-L3 and SMOS-L3 are the low-frequency pass microwave products that provide a composite of daily estimates of global land surface soil moisture retrieved by the L-band at 9-km and 25-km resolution,

respectively. LPRM_AMSR2_DS_A_SOILM3 (denoted as AMSR2) is a high-frequency pass microwave product, and we use the X-band data to estimate global soil moisture (de Jeu, Richard, 2013; Owe et al., 2008). GLDAS_NOAH025 integrates ground-based observation data and satellite data to drive land surface models to estimate hydrologic variables including soil moisture. We also compared the multitask model to another machine learning-based model, SoMo.ml, obtained by an LSTM model trained solely on global in-situ networks (O and

Orth, 2021). This model has been evaluated on global in-situ networks using the spatial cross-validation method. Notably, SoMo.ml product provides soil moisture estimation from 0-10 cm depth, not 0-5cm depth. Its final product was obtained by retraining the model using all available sites and times rather than by using spatial cross-validation. All of the comparison datasets and results are listed in Table S3 and Table S4 in the Supplementary Material, respectively.

### 180   2.4. The experiments.

To understand the model's performance for short-distance spatial interpolation, we ran random K-fold cross-validation for random spatial tests. To understand performance for long-distance spatial extrapolation, we slightly modified this procedure and ran cross-continental tests. In K-fold cross-validation, as in our multi-scale process (Liu et al., 2022), we randomly separated the in-situ and satellite data into 5 groups. In each round, we used 4 of the 5 groups to train the multitask model, and used the remaining one for testing. We repeated this for 5 rounds,

so that each point was tested. In the cross-continental test, we divided the global data into 7 large regions. In each round, we kept one region's data as the test set and used the rest as the training data set. We repeated this process





7 times, so that each region was treated as the test region once. Both the spatial training and test periods were from 01 April 2015 to 31 December 2020. We also ran temporal tests, for which the training period was from 01 April 2016 to 31 December 2020, and the test period was from 01 April 2015 to 31 March 2016.

**2.5. Analysis of factor controls.**

We used a stratified analysis to explain which variables may have had control over the model's performance. We first trained a random forest (RF) model from the sklearn library (Pedregosa et al., 2011) in order to identify the first few important factors of the LSTM model influencing correlation (R) in either temporal or spatial tests. Briefly, RF uses a collection of decision trees to predict the R values. At the nodes of each tree, the data is split into two bins to minimize the variance of the bins after the split. Therefore we could calculate the average contribution of each factor to the reduction of variance and then obtain the ranking of importance. Note that the importance ranking is not about "*is factor A important for predicting soil moisture?*", but rather "*are there certain ranges of a factor, or joint ranges of multiple factors, where the model behaves more poorly than other ranges?*". From the importance results, we chose two importance factors and plotted R as a function of these factors to explore and interpret how they controlled model performance. The goal was to gain a physical interpretation of why the model sometimes produced lower-quality outputs, and offer some possible guidance about when to be more cautious in relying on model results.

**3. Results and Discussion**

**3.1. Error types and temporal tests.**

Before we dive into the results, we first need to discuss several error types so it is easier to interpret the results. We can roughly separate soil moisture modeling errors into multiple components: (A) climatic forcing errors; (B) training data limitations and nonstationarity (e.g., the model being unable to learn the correct response to drastic changes that have never been seen before); (C) errors due to uncaptured spatial heterogeneity in soil properties; and (D) model training errors (i.e., overfitting or underfitting to the training data resulting in mismatches for the testing data). Among these, A and B are likely to manifest as errors in the temporal tests. B especially appears as large temporal test errors when compared to the spatial test errors, which would indicate strong nonstationarity. Both B and C can be reduced when there are more numerous or more accurate training data. C appears as a large error in the spatial test, indicating that either the available soil property data are not accurate or diverse enough to reflect the impacts of soil texture, or there are local hydrologic processes, e.g,. riverine inundation or irrigation, that are unknown to the LSTM (not contained in the inputs). C will also modestly decrease as data density increases (as training sites inherently become closer together), but typically cannot be removed entirely. D appears as a large difference between training and testing metrics. It is worthwhile to note that due to a "data synergy" effect (Fang et al., 2022), LSTM models typically (although not always) perform better on each site when given data from more numerous or more diverse sites.

The temporal tests (trained on some sites in one time period and tested on the same sites in another time period), which are used to establish a reference performance level, showed a strong ability for LSTM to capture soil moisture dynamics around the world, with a global median correlation (R) of 0.837 (Table 1a & the sky blue,



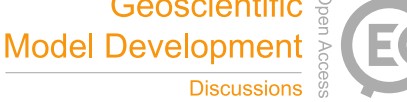

second box from the right in Figures 1 & 2). Because LSTM has learned from the history of the sites, these test
region-aggregated metrics are normally higher than spatial tests (except for Alaska, which is to be discussed
below), and reflect the inherent and geographically-varying difficulties of soil moisture modeling in different
regions. The R values for different regions follow this order:
Africa_North>Australia>Asia>CONUS>Europe>Africa_South ~=Alaska. One immediately apparent
observation is that this order is not related to the number of sites in each region, nor the density of sites. For
example, the highest-ranking (in terms of R) regions are Africa_North, Australia, and Asia, which all are among
the regions with the lowest counts of sites. Alaska has a relatively high site density, but had the lowest median R,
which could be attributed to the unique difficulties associated with frozen soil and thawing permafrost. This
observation suggests that more training sites on these continents may not result in significantly better temporal
test results at existing sites. Africa_South was more difficult than Africa_North, presumably because more sites
are located in arid environments (LSTM has previously shown lower performance in such regions in the CONUS,
as discussed in Feng et al. (2020)). While these results show that there are some regions in the world that are more
difficult to capture than others for the prediction of soil moisture, the overall results are encouraging. The model's
performance over these regions indicate that the quality of the forcing (MSWEP+GPM+ERA5 precipitation) and
soil characterization data is globally consistent.

Apart from Alaska, there were no particularly strong spatial patterns in either R or RMSE in the random spatial
(cross-validation) tests (Figure 3). Over the CONUS, there was a mild concentration of poor performing sites in
the northwest. In Europe, we found poor-performing sites in the central region, e.g., Hungary and Romania. Other
than that, poorly performing sites were interspersed among the well-performing sites, suggesting that most of the
causes of poor performance are local rather than climatic effects, which we will explore in Section 3.3. The cross-
continental tests led to a widespread decrease in R, in comparison to the random spatial tests (Figure 4). While
some African sites, like those immediately south of the Sahara desert (Figure 4c), had noticeably deteriorated
performance, some other sites in fact improved, like the most southern three sites in Africa_South (Figure 4f).

**3.2. Randomly-sampled spatial cross-validation**

The random spatial (randomly-sampled cross-validation tests), which examined the effect of spatial interpolation,
showed record-breaking results despite their slight performance decline compared to the temporal tests. The global
median R was 0.792, ubRMSE was 0.056, and RMSE was 0.075, all of which were slightly better than the CONUS
median values (Table 1b & the wheat-colored, third box from the right in Figures 1 & 2). In contrast, the SMAP
9-km product and GLDAS had global median R values of 0.621 and 0.699, respectively. As expected, at the global
scale, the correlation for training the multitask model was slightly higher and had a smaller spread than that for
the temporal and spatial tests. It should be noted that the numbers are not entirely comparable: SMAP 9-km and
GLDAS were not calibrated fully on the sparse in-situ sites. The LSTM-based SoMo.ml model obtained a median
R of ~0.6 for spatial cross-validation (Figure 7 in O & Orth, 2021), while the downloadable SoMo.ml product
(0.805 as shown in Figures 1 & 2) was obtained based on training on all the sites and time periods and thus should
in fact be compared to the multitask model training period results (the rightmost box in each panel, R=0.853). It
should be noted that SoMo.ml has soil moisture for multiple depths but we only explored the 5cm product here.
The closest model to the multitask LSTM is the one from Beck et al. (2021) (we do not have the data to plot their



results), which was calibrated on 177 of the soil moisture sites and tested on the others. Their MSWEP+HBV
model obtained a median R value of 0.78. Their performance is competitive and quite impressive for a process-
based model, but unfortunately the HBV model only outputs a water storage value (in mm) that can be correlated
to the fluctuation of observed soil moisture, not the soil moisture itself, and thus other metrics like bias cannot be
calculated (additional linear transformations are required to obtain soil moisture, which introduces uncertainty).
It would be interesting to explore how HBV or similar models would react to the cross-continental test below,
where it may show some advantages.

In general, the difference between the training and temporal test is small so we regard the model training error to
be small. Switching from the temporal test to the random spatial test, most regions suffered a small decline in
performance, suggesting the impact of spatial heterogeneity is larger than the impact of temporal nonstationarity
for soil moisture predictions. Regions seeing noticeable declines include the CONUS (from 0.847 to 0.790), Asia
(0.873 to 0.762), and Australia (0.877 to 0.778), which could reflect the limited quality of soil texture data and
processes that cannot be described by the input attributes. Alaska stood out as the exception (temporal test
R=0.654, spatial test R=0.789), which is in fact consistent with our theory of errors and highlights the rapid
changes facing arctic regions. Alaska is challenging because it is the frontier of rapid changes and is experiencing
rapid permafrost thawing and changes in frozen ground months. As a result, temporal nonstationarity there trumps
spatial heterogeneity. This observation suggests that the soil moisture dynamics in Arctic regions in the coming
years will differ materially from those in the past decades.

Precipitation data quality exerts an influence on the performance of the model but does not materially change the
model comparisons. MSWEP is a high-quality global precipitation dataset (daily, 0.1° resolution) arising from
blending multiple forcings datasets and correcting their biases (Beck et al., 2019). To support a fair comparison,
we also ran our multitask model with the more widely used ERA5 precipitation data, which gives slightly lower-
performing results but still outperforms comparison products (Table S5 in Supplementary Material). We further
note that our previous CONUS results used the NLDAS forcing data, which is more customized toward North
America, and obtained an R of 0.901(Liu et al., 2022). We thus conclude that the forcing dataset used has a
moderate impact on results, and needs to be the same for models to be fully comparable.

### 3.3. Cross-continental tests.

As expected, model performances dropped significantly in the cross-continental test (testing on a continent where
no training data was provided), but even under this adverse situation, the multitask LSTM model surpassed or
equaled the performance of SMAP in all regions except Asia (Table 1c, Table S4, Figures 1 & 2). When the
CONUS was included as a training region, the R value in all regions except Alaska stayed above 0.64. When both
the CONUS and Europe were included (again, except for Alaska), there seemed to be a baseline performance
(R=0.70) which the model would not fall below, despite there being no training data from the test continent. For
Africa and Australia, the advantages of the multitask model (multitask_exclude in Figure 2) over SMAP or
GLDAS are prominent. This suggests even in highly data-sparse regions, we could consider the multitask LSTM
model to be a viable product.

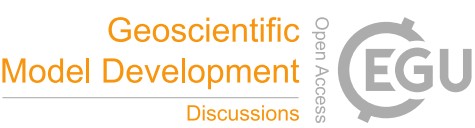

Interestingly, the fewer sites a region had, the less impact there was by switching from the random spatial to the cross-continental test (Table 1c & third box from the left in Figures 1 & 2). For Africa_North, Africa_South, and
Australia, there was no decline from the random spatial test (Table 1b) to the cross-continental test (Table 1c). Asia saw a larger impact, with R dropping from 0.762 in the spatial test to 0.711 in the cross-continental test. We notice precipitous drops for Alaska (median R from 0.789 to 0.581 -- again suggesting soil moisture dynamics there are materially different from other parts of the world), Europe (0.791 to 0.646), and the CONUS (0.790 to 0.605). We thus conclude that when a region had very few sites but high heterogeneity, these sites only played a
minor role in training the model, and thus removing them did not materially change the model. When a region had a large number of sites, like the CONUS or Europe, removing them substantially reduced the training data diversity. The quality of a DL model is a strong function of its training data -- thus it would be severely weakened if a large part of its training data were removed.

It is also interesting that African, Australian, and Asian sites had good performance in this experiment. It seems to suggest their soil types and rainfall-moisture responses have already been covered by similar sites in the CONUS and Europe, and thus the model was already sufficient. With diverse climates and landscapes ranging from desert to temperate forest and from croplands to wetlands, the CONUS networks play a dominant role in providing training on how soil moisture responds to different forcings, as modulated by the soil and landscape
characteristics. We cannot know for certain that the model will work well in other parts of Africa and Asia until we have more in-situ sites there. However, the current results at least can make us hopeful that, when trained on all globally available data, the model will likely produce good results in some parts of the untrained world and will likely add value beyond satellite products.

### 3.4. Factorial influences on model performance

Due to LSTM's strong ability to fit to data, it can serve as a probe for process complexity (Liu et al., 2022; Feng et al., 2022; Tsai et al., 2020; Feng et al., 2020): those sites that LSTM cannot adequately capture may contain complicated processes that are not well described by the inputs. The factorial importance analysis indicates that slope aspect, average soil moisture, and surface solar radiation downwards are the top three factors that influence the multitask LSTM model's R in the temporal test (Figure 5). As a reminder, this importance test is based on
training random forest (RF) models with these inputs listed in Figure 5, and R from either temporal or spatial tests as the targets. A high-ranking factor in Figure 5 implies that it not only has influence on soil moisture, but also on the predictability of soil moisture. It could be that in a certain range of this factor (the range may be conditional on other factors due to factorial interactions) there are not that many sites of this kind (it is a minority class that is not well represented in the training dataset), or that some latent processes become important. Nevertheless, due to
the inherent limitations of machine learning, factorial importance is only a hypothesis rather than confirmed truth (Tsai et al., 2020). As a result, human interpretation of the results will be required. Because the sensitivity to radiation is somewhat difficult to interpret, here we focus on aspect and average soil moisture.

The model correlation in the temporal test generally rises as soil moisture goes up, until reaching the wettest
regime (0.48-0.6), where its variability increases (Figure 6-I-h). The sites in the middle range tend to have continuity in soil moisture and regular rainfall patterns, which are most ideal for LSTM. The driest sites may be

difficult to predict due to scarce but sudden rainfall events that quickly dry out, which reduces the usefulness of LSTM's memory capability. Using the spatial test R as the target (Figure 6-II-h), the pattern is similar but less pronounced, which suggests the driest sites are also more impacted by temporal non-stationarity than spatial

heterogeneity, because they have seen limited storm events. Toward the wettest regime, saturation often occurs, and soil moisture may be influenced by groundwater processes which are difficult to account for.

Interestingly, aspect has a nonlinear effect that varies in different soil moisture regimes (Figure 6-I-i) due to its impact on shading and solar insolation. It is well known that aspect can have a predominant control on soil

moisture and plants for dry sites, as witnessed by different vegetation densities and species and microbial communities on south-facing and north-facing slopes (Armesto and Martínez, 1978; Bennie et al., 2006; Xue et al., 2018). For the very dry sites (average SMAP<0.08), only those with mid-range aspects tended to have a decent correlation. The temporal test R (Figure 6-I-i) had a larger response to aspect than the spatial test R (Figure 6-II-i), which suggests this difficulty is not a result of too few training sites in space, but a result of highly complex

and nonstationary temporal trends in this combined range of average soil moisture and aspect. The north-facing dry slopes have a lower R perhaps because of complex vegetation-soil moisture interactions in this regime, which may shift from year to year. The most south-facing dry slopes also have low R, perhaps because they approach the lower limit of soil moisture and can see large changes due to individual storm events. On the other hand, for the wetter soil regimes, the role of aspect is reduced --- we see noticeably reduced R only for the most south-

facing slope (Figure 6-II-i). This reduced impact may be because soil moisture is not such a strong selector of vegetation species on these slopes and thus the distinction of aspect becomes less important.

In the vast parts of Africa or Asia where soil moisture predictions are required but not well-supported by in-situ measurements, the analysis above can help us to anticipate challenges. At the hillslope scale, our predictions may

have a larger error for those north-facing slopes in the dry regime and also straight south-facing slopes for the Northern Hemisphere (to be reversed for the Southern Hemisphere). The results highlight the importance of aspect controls on soil moisture and suggest that future models will need to well represent its effect before they can be accurate.

### 3.5. Further discussion

Our correlation is modestly higher than the previous state-of-the-art model, the well-calibrated conceptual hydrologic model, HBV. Even though that model does not simulate the physical quantity of soil moisture, it could be modified to have a module that does. However, to obtain suitable parameters on the global scale and improve the physical processes, we think adding differentiable programming to the model will give it the adaptive capability to learn from big data (Feng et al., 2022). It is possible that such a model may generalize better than

LSTM over long distances due to the imposed physical constraints.

Typically, for many hydrologic applications (Fang et al., 2022; Feng et al., 2021; Liu et al., 2022; Rahmani et al., 2021a), a spatial test is a tougher test than a temporal test for fully data-driven models, showing the strong impacts of spatial heterogeneity. This could either mean the inputs of the model do not completely describe the problem,

or there are not enough sites in space with different combinations of input attributes for the model to fully resolve



their impacts. Typically, spatial error can be gradually reduced if there are more training sites in space. However, in both Alaska (Figure 1) and north-facing dry slopes around the world (Figure 6-II-i), temporal errors have exceeded spatial errors. Consistently, when we ran K-fold experiments with a higher K, it also did not result in noticeably different performances for the models (data not shown). These observations not only highlight the

unique challenges of these places (rapid climate-driven changes and strong nonstationarity), but also suggest that the number of training sites is not a predominant issue for limiting the accuracy of soil moisture predictions.

While our product, under the most stringent test (cross-continental), did not surpass SMAP by a large margin, it is suitable as a long-term simulation tool, as it does not require near-real-time observations. Thus it can be used

to assess future climate change impacts. It is also easy to further expand LSTM networks to enable "data integration" or "data assimilation", which absorbs information from recent observations to improve future forecasts (Fang and Shen, 2020; Feng et al., 2020). Satellite observations could also be employed as the recent observations, as it could help to update LSTM's hidden states. Such assimilation typically results in a significant boost in performance and the elimination of bias. Compared to data assimilation with traditional models, we could

skip the bias correction procedure, as LSTM models tend to have little bias and will adaptively learn to remove the bias by themselves. Data assimilation only has short-term impacts, however, and the value of the information content of the data will eventually wane as the simulation proceeds.

The LSTM-based SMAP modeling product is already deployed at scale via the operational agricultural advice

application of PlantVillage, a nonprofit organization based at Penn State. We intend to put the multitask model into production alongside alternative estimates. This service is provided free-of-charge to farmers and extension services in Africa through the USAID Current and Emerging Threats to Crops Innovation Lab (CETC IL). PlantVillage currently scales out precipitation data to 13 million farmers/week in Kenya and Burkina Faso, and believes the ability to complement this with more accurate information on soil moisture will be of large assistance

to farmers coping with droughts and erratic weather as a result of climate change. It is also valuable to help farmers optimize fertilizer application rates, which has become even more critical due to the massive increase in fertilizer prices over the last 12 months.

## 4. Conclusions

When evaluated against sparse in-situ soil moisture networks, the multitask LSTM model outperformed currently

available satellite-based products, land surface models, and an alternative DL model, across most continents. Judging by the 5-fold spatial test model results, the model not only had dramatically lower bias but also the highest correlation with in-situ soil moisture networks. Learning from multiple data sources, the model can be deployed at large scales at a small computational cost, and can be expanded to incorporate data assimilation capabilities. These features make it a suitable operational tool to democratize access to information for agriculture in

developing regions. While we wish for more measurements in Africa for model training and validation, the results are at least encouraging. The model can utilize satellite-estimated soil moisture as one of the learning targets while also learning from in-situ data, and thus is well-poised to provide higher-resolution outputs than the satellite-based products.





The LSTM model served as a probe for process complexity and showed that mean soil moisture and aspects have important controls on soil moisture predictability, while Arctic regions are inherently more difficult due to rapid soil changes. For the dry slopes (average SMAP soil moisture <0.08) that face north, there could be complicated vegetation-soil moisture interactions that are difficult to predict. For the wetter slopes, the role of aspect becomes less prominent. Error analysis suggests that in these difficult regions, temporal errors can outweigh spatial errors,

thus having longer data records and monitoring most recent changes can be more important than adding more sites.

The multitask LSTM model can generalize well into highly data-sparse regions. Even in the worst-case scenario (no training data on a whole continent), the model was able to surpass SMAP's accuracy on most continents. It

did seem to have some trouble generalizing to Alaska, where the soil dynamics are much different from other regions and are also experiencing rapid changes. However, it provided decent performance when tested in data-sparse continents where it has not been trained, like Africa and Australia, showing that these predictions can be beneficial for such regions where there are not a lot of published soil moisture datasets. This modeling success is partially due to the strong ability of the model to generalize, but also because the soils in the known sites in Africa

are similar to those in the training set. It is fortunate that the more intensively instrumented CONUS and Europe already contain a wide variety of soils and climates for training, without which the model would suffer greatly.

## 5. Code/Data Availability

The multitask LSTM code and GSM3 soil moisture dataset can be downloaded at https://doi.org/10.5281/zenodo.7026036. Links to data sources have been provided in the Methods section.

**6. Author Contribution**

C.S. conceived the study; J.L. ran the experiments and wrote an early draft; J.L., D.H., F.R., K.L., and C.S. edited the manuscript.

## 7. Competing Interests

Chaopeng Shen and Kathryn Lawson have financial interests in HydroSapient, Inc., a company that could

potentially benefit from the results of this research. This interest has been reviewed by the University in accordance with its Individual Conflict of Interest policy, for the purpose of maintaining the objectivity and integrity of research at The Pennsylvania State University.

## 8. Acknowledgments

This work was supported by Google.org's AI Impacts Challenge Grant 1904-57775 and Gates Foundation award

INV-018429. Shen was partially supported by National Science Foundation Award OAC #1940190.



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



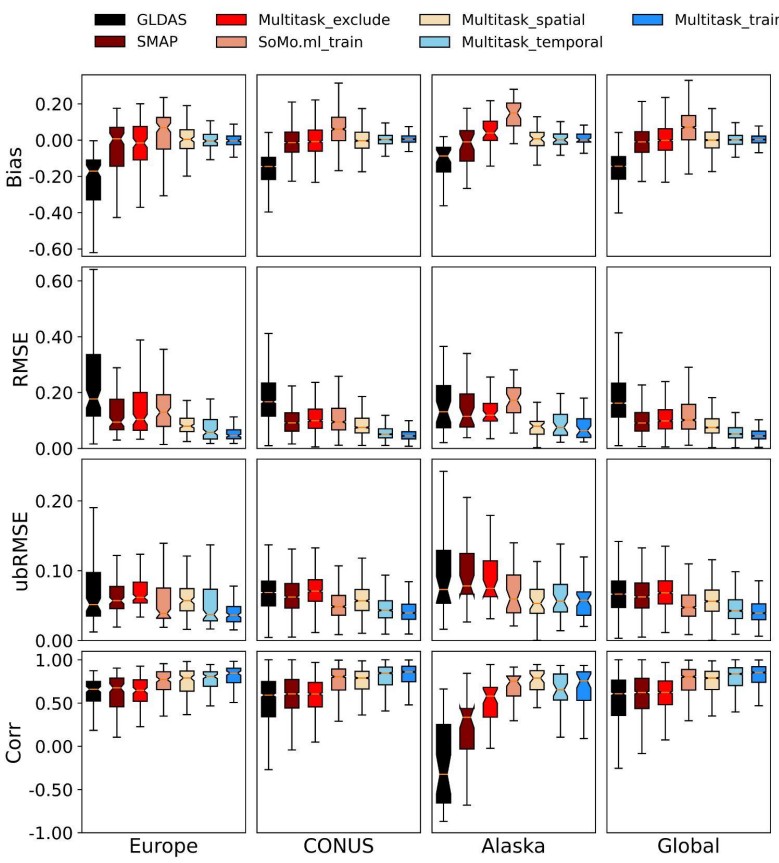

**Figure 1. Comparison of model performances for different continents in data-rich regions. Models from left to right are ranked from lowest to highest global correlation. We plotted results for the training period as well as temporal, spatial, and cross-continental tests. "Multitask_exclude" means the cross-continent test: the models were tested on a continent but sites from that continent were excluded from training. The SoMo.ml product shown here was trained on all sites in all time periods so it is most comparable to our "Multitask_train" product.**



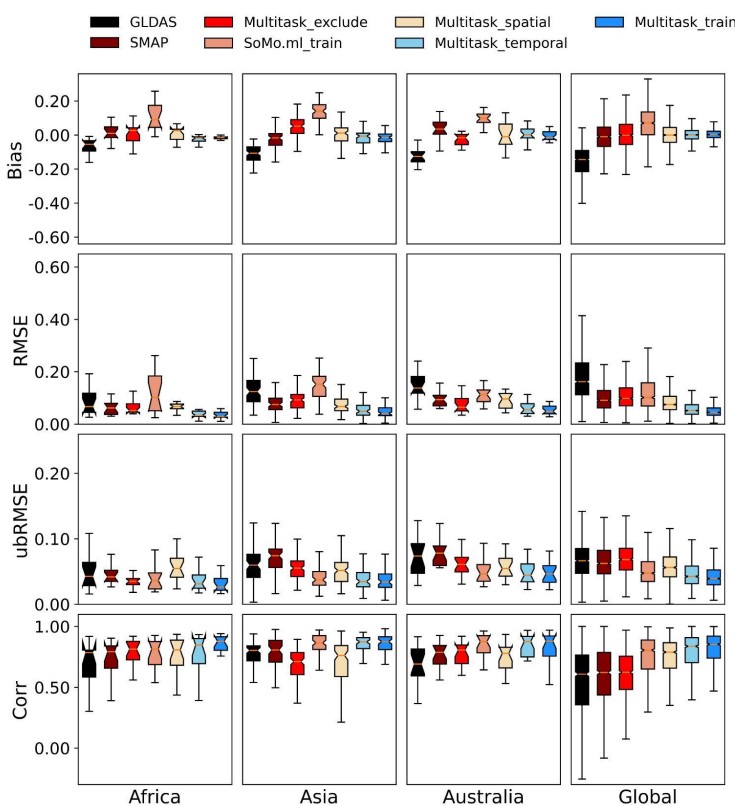

**Figure 2. Same as Figure 1 but for data- sparse regions: Africa, Asia, Australia.**






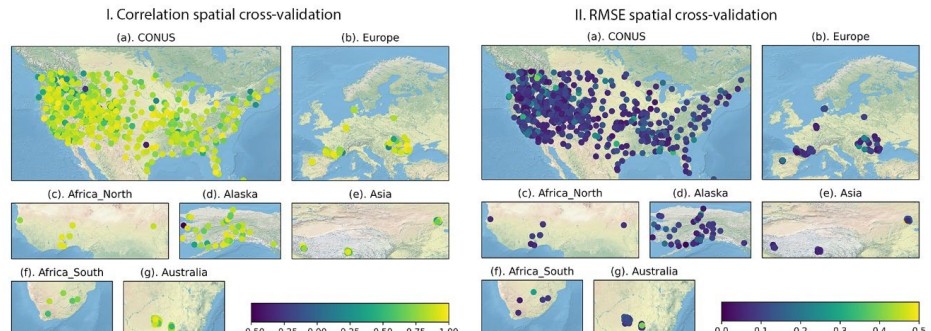

**Figure 3. Metric distributions for the multitask model random spatial cross-validation tests.** (I.) Correlation and (II.) RMSE of spatial cross-validation tests for (a) the CONUS, (b) Europe, (c) Africa_North, (d) Alaska, (e) Asia, (f) Africa_South, and (g) Australia. The training and testing period were both from April 1, 2015 to December 31, 2020. Maps are made with Natural Earth imagery, no permission needed (Made with Natural Earth, 2022).




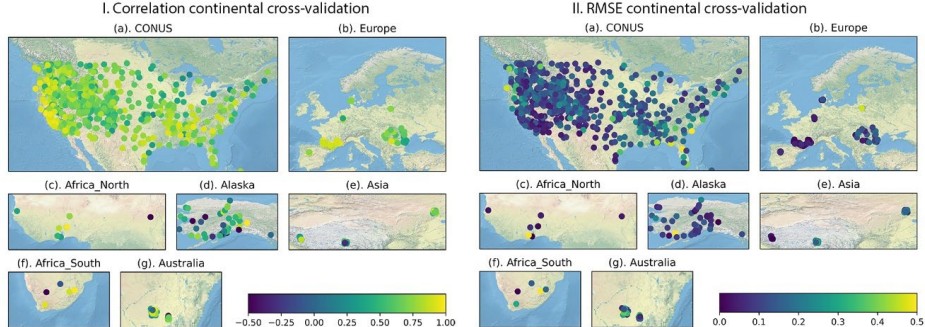

**Figure 4. Metric distributions for the multitask model continental cross-validation tests. (I.) Correlation and (II.) RMSE of continental cross-validation tests for (a) the CONUS, (b) Europe, (c) Africa_North, (d) Alaska, (e) Asia, (f) Africa_South, and (g) Australia. The training and testing period were both from April 1, 2015 to December 31, 2020. Maps are made with Natural Earth imagery, no permission needed (Made with Natural Earth, 2022).**




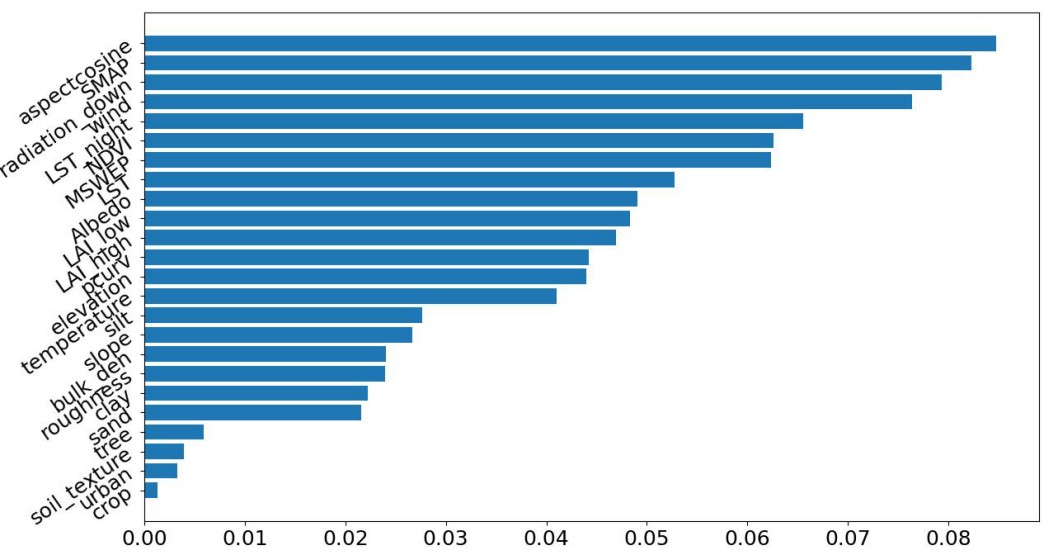

**Figure 5. The importance of the features in the RF model constructed using all the unduplicated category**
**data presented in this paper.**



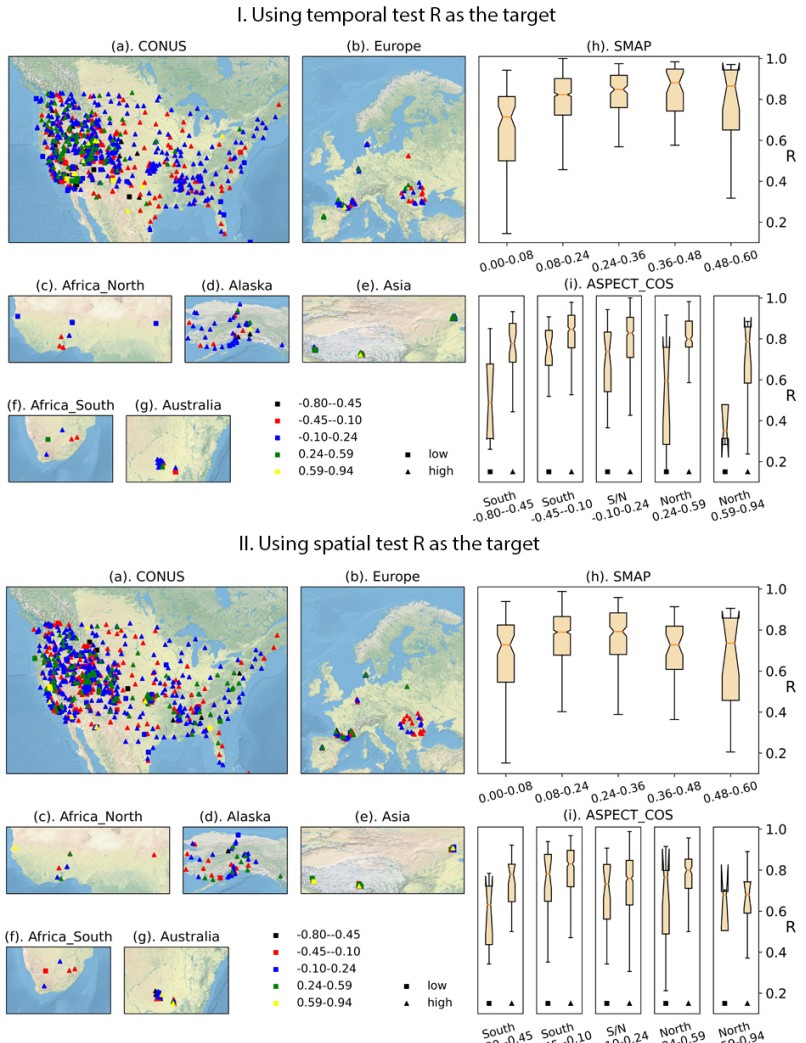

**Figure 6. Stratified analysis of the distribution of R values from (I.) temporal and (II.) spatial tests. (a-g) The maps show the global distribution of test sites as a function of average SMAP soil moisture value and aspect. The colors on the map represent aspect cosine. The average SMAP<0.08 sites are a minority class and are represented by squares. (h) The SMAP boxplot shows the distribution of R under different average soil moisture values (SMAP). (i) The aspect boxplot shows the distribution of R in different aspect cosine bins, where the left one indicates SMAP <=0.08, and the right one indicates SMAP>0.08. The upper panels show temporal test R values (which characterize temporal nonstationarity) while the lower panels show spatial test R values, which characterize the effect of spatial heterogeneity. Maps are made with Natural Earth imagery, no permission needed (Made with Natural Earth, 2022).**



**Table 1. Model's performance in three scenarios. (a) The model's temporal testing in different regions. (b) The model's spatial cross-validation testing in different regions. (c) The model's continental cross-validation testing in different regions.**

| | | | | | | | | |
|---|---|---|---|---|---|---|---|---|
| **(a) temporal testing** | | | | | | | | |
| Median metrics | CONUS | Europe | Africa_North | Alaska | Asia | Africa_South | Australia | Global |
| Bias | 0.003 | -0.005 | -0.014 | 0.001 | -0.007 | -0.044 | 0.001 | 0.001 |
| RMSE | 0.051 | 0.058 | 0.031 | 0.075 | 0.049 | 0.056 | 0.055 | 0.051 |
| ubRMSE | 0.043 | 0.037 | 0.026 | 0.056 | 0.035 | 0.048 | 0.044 | 0.043 |
| Corr | 0.847 | 0.808 | 0.881 | 0.654 | 0.873 | 0.656 | 0.877 | 0.837 |
| **(b) spatial cross-validation testing** | | | | | | | | |
| Bias | -0.004 | 0.004 | 0.029 | 0.007 | 0.011 | 0.021 | -0.010 | -0.0003 |
| RMSE | 0.075 | 0.080 | 0.067 | 0.079 | 0.067 | 0.074 | 0.096 | 0.075 |
| ubRMSE | 0.057 | 0.057 | 0.048 | 0.053 | 0.052 | 0.071 | 0.055 | 0.056 |
| Corr | 0.790 | 0.791 | 0.861 | 0.789 | 0.762 | 0.647 | 0.778 | 0.792 |
| **(c) continental cross-validation** | | | | | | | | |
| Bias | -0.009 | -0.016 | 0.041 | 0.039 | 0.052 | -0.067 | -0.016 | -0.002 |
| RMSE | 0.099 | 0.104 | 0.047 | 0.119 | 0.092 | 0.078 | 0.065 | 0.098 |
| ubRMSE | 0.071 | 0.062 | 0.032 | 0.075 | 0.055 | 0.052 | 0.061 | 0.068 |
| Corr | 0.605 | 0.646 | 0.87 | 0.581 | 0.711 | 0.718 | 0.806 | 0.624 |