# Peer review of "Evaluating a Global Soil Moisture dataset from a Multitask Model (GSM3 v1.0) for current and emerging threats to crops"

_Geoscientific Model Development, 2022_

## Author Comment (AC1)

*The manuscript "Evaluating a Global Soil Moisture dataset from a Multitask Model (GSM3 v1.0) for current and emerging threats to crops" presents a LSTM-based machine learning model for global soil moisture estimation. The authors used a multi-loss framework to train the LSTM model, and employed multiple reference datasets for evaluation. Specifically, the work investigated spatial generalization ability by cross-validations, both randomly and continent-based sampling. The overall quality of the work is excellent and fits in the scope of GMD. With that, I do have the following comments.*
Thank you for your evaluation.

*The title is somewhat confusing to me. It indicates the authors also care about crops in addition to soil moisture, as soil moisture will affect the crops. So I was expecting to read something about agricultural applications (e.g., use the DL-based soil moisture to drive some crop models). However in the manuscript, it is only mentioned before the conclusion section that this model will be put into production, and the major focus of the paper is model benchmarks on soil moisture. I suggest the authors have more discussion on crop applications, or revise the title to make it clearer. In addition, if the main focus is to evaluate the dataset instead of the LSTM model, I suggest making the dataset publicly available and adding a link to access this dataset.*

We understand the title may be a little confusing, but it is a bit difficult to change. The overall motivation of this work, and the way it will be used, is to support the detection and prediction of threats (like pests) to crops in a region like Africa so we hope the mission is reflected in the title. However, any other option we've come up so far are either too long or misses the mission or other key elements. Any suggestion would be welcomed.

Actually, we have provided the code and data link in the "Code/Data Availability section".
https://zenodo.org/record/7105958#.Y2SVjHbMJGY

*To clarify, what's the difference between the SoMo.ml model and the authors' LSTM model? Do they share the same dynamic and static input variables with the only difference as the loss function? In Figure 1 and 2, the authors compared the metrics derived from the training period (Multitask_train and SoMo.ml_train). The R value or RMSE from the whole training period is not important, as one can always overfit the model. I understand that a barplot for Multitask_train shows the model is not overfitting, but a comparison between Multitask_temporal and SoMo.ml_temporal would make more sense to me.*

There are three major differences between SoMo.ml and multi-task models:
1. SoMo.ml used 13 different types of dynamic and static data as input data, while multi-task used 35 different data types, and their data sources and resolutions are also different.
2. The model's loss calculation is different. SoMo.ml trained the model using ~1000 sites worldwide, so the loss is calculated based on the in-situ data. The multi-task model trained the model with both in-situ and satellite grid data, and the loss is calculated based on both in-situ and satellite data.
3. The model structure is different too. SoMo.ml uses the input data from day t-364 to day t to obtain the target soil moisture data on day t, which is a sequence-to-one structure. The multi-task model uses the input data from day t-364 to day t to obtain the target soil moisture data from day t-364 to day t. It is a sequence-to-sequence structure.

We will add the following to the paper:

*Line: 180. "Besides, the SoMo.ml model differs from the multi-task model in terms of input data, loss value calculation, and model structure. However, it is still helpful for us to understand better the performance of different soil moisture products by comparing the final products."*

SoMo.ml only provided the final products which used all data. We can not access their temporal test results and can not make the comparison. More importantly, their model was retrained using all available data, so they are the result of the training period while we are in the testing period. Even in this unfavorable situation, we still achieved a good performance. This question has been answered in the paper:

*Line: 177. "Notably, SoMo.ml product provides soil moisture estimation from 0-10 cm depth, not 0-5cm depth. Its final product was obtained by retraining the model using all available sites and times rather than by using spatial cross-validation"*

*What is ubRMSE in Table1? Does Corr represent Pearsonr correlation coefficient? In Figure 3 and 4, the meanings of colormaps are not the same. In the correlation map, greener represents better performance, but it is worse performance in the RMSE plot. I would suggest a uniform colorbar with light (dark) colors for high performance and dark (light) colors for low performance.*

ubRMSE is unbiased RMSE. Each time series was removed of its mean before they are compared to the observations, which also have their means removed. We will add the "2.6 Evaluation Metrics" section to explain the meaning of each metric.

*Line 218: "The metrics used to evaluate the Multitask model's performance include Pearson correlation coefficient (Corr), Bias, Root-mean-square deviation (RMSE), and unbiased RMSE (ubRMSE), in which RMSE is calculated after bias is removed. These metrics are the median value of all satellite grids and in-situ. When we calculate these metrics, we remove the observed and predicted data when there is nan in the observation."*

We will change the colorbar in Figure3, Figure4, and Figure S1, with dark colors for high performance and light colors for low performance.

*The LSTM model is optimized towards both the in-situ estimation and the satellite products. In Figure1 when comparing the model performance, the authors selected the SMAP and GLDAS products, which are gridded datasets. When evaluated against the ISMN dataset, it is not a fair comparison because of the coarse resolution of SMAP. How do the authors correlate in-situ measurements with gridded datasets? Will the result change when switching to 25-km resolution (i.e., with coarser resolution, the dataset will lose more representations)? A further question is that, what is the meaning of the LSTM outputs? Is it the best estimation over the grid points (e.g., an average over the grid spacing), or the best estimation for the in-situ observations?*

We are not really attempting to outcompete SMAP which is an observation and are used as a training data. It would be impossible to compare to "SMAP or GLDAS optimized for comparing to sparse in-situ data"

because such products do not exist. However, we think the community would want this comparison in the Figure to serve as a helpful context to know where our model stands, so we chose to put them in. To clarify this, we will add the following:

Line 174: "*It is to be noted that SMAP and GLDAS products were not optimized to match the sparse networks so this comparison is not entirely fair, but was shown to provide a context.*"

The meaning of the LSTM product is the average soil moisture for the 9-km grid. The multi-task model input data include satellite grid data (9-km) and in-situ data, where the in-situ forcing, and static attribute data are extracted for the 9-km satellite grid. The model's prediction resolution is 9-km, so it is fair to directly compare the performance of SMAP and the model at the ISMN location. However, the GLDAS's resolution is 0.25 degrees, Corr is 0.609, Bias is 0.045, RMSE is 0.101, and ubRMSE is 0.069. For a fair comparison, we adjusted the multi-task model's resolution to 0.25 degrees and resampled the other products to 0.25 degrees (Table S5). Therefore, the model's performance gets slightly worse as the resolution gets coarser, but it does not change our conclusions.

Line 185: "*We also resampled the model's input data and the other products. They were compared at the same resolution of 0.25 degrees. The model's performance dropped slightly but with the same conclusions as the 9-km resolution (Table S5).*"

We trained the multi-task model using the satellite grid and in-situ data. The model is still an LSTM model, but its loss values is calculated from the satellite loss and in-situ loss. Therefore, the LSTM model's output depends on its input when predicting soil moisture. If the input data is satellite grid data, then the output is also grid soil moisture prediction. If the input data is in-situ, the output is in-situ location soil moisture prediction. In this paper, we used in-situ input to obtain the soil moisture prediction to compare with ISMN. We used global grid data for the final products to get the global 9-km soil moisture from 2015 to 2020. We will add the following to the paper:

Line 183: "*The model performance under different experiments is compared with the IMSN in-situ data, while the final product input and output data are both global 9-km grid data.*"

*The authors split the global dataset into 7 continents. Would it be a more straightforward comparison to have a similar 7-fold cross validation to match the number of continents in Section 3.2? I believe 5-fold is a common choice but 7-fold may be a more intuitive and fair comparison.*

In the paper, we used a 5-fold method for the global data to get the results: Corr is 0.792, Bias is -0.0003, RMSE is 0.075, and ubRMSE is 0.056. When we trained the model using 7-fold, Corr is 0.797, Bias is -0.0004, RMSE is 0.075, and ubRMSE is 0.056. The results did not differ significantly. We will add the following to the paper:

Line 191: "*In cross-validation, 5-fold or 10-fold is a common choice, and we can also use 7-fold according to the number of regions in the spatial cross-continental. However, after testing, there is no significant difference in their results. To save computational resources, we used 5-fold for all experiments.*"

*When analyzing the factor controls, the authors selected a random forest model. What is the performance/skill of this random forest model? Is Figure 5 using spatial R as target or temporal R, or is it a multi-output model? At line 329 it shows the target is temporal R, but line 330 shows R is from either temporal or spatial tests.*

Random Forest is a classification/regression algorithm consisting of many decision trees that use bagging and randomness of features to create a series of decision trees. It is suitable for non-linear data and reduces the risk of over-fitting. We want to make the paper simple, so we chose the RF model that is easy to implement and performs well. To understand when the model performs better, we use the random forest method to rank the important factors. RF calculates each tree's mean and standard deviation of impurity reduction accumulation to get interpretable explanations.

*Line 204: "Random Forest (RF) model is a classification/regression algorithm consisting of many decision trees that use bagging and randomness of features to create a series of decision trees. It is suitable for non-linear data and reduces the risk of over-fitting."*

For Figure 5, we only used temporal experiment R, so we modified the caption of Figure 5 to *"In the temporal experiments, the importance of the features in the RF model constructed using all the unduplicated category data presented in this paper"*. We also modified *"... and R from either temporal or spatial tests as the targets."* to *"... and R from temporal test as the targets."*

*Regarding the random forest model, is the conclusion independent of the model choice (or what's the reason to choose a random forest model)? Will we get different results if we switch to extreme boosted trees or other tree-based models? Also please check the name of the python package. It is used as "sklearn" in python but the paper referenced (Pedregosa et al., 2011) shows "scikit-learn" in the title.*

Different models use different algorithms so the results may be slightly different. We built Gradient Boosted Decision Trees (GBDT) to analyze the important factors. "Aspectcosine", "MSWEP" (precipitation), and "Downward shortwave radiation" are the top three ranking factors in the results. Compared with Random Forest, precipitation ranked higher than SMAP in Boosted Trees, but this does not affect our conclusions. Because the precipitation and soil moisture trends are the same, they are equally represented. We will add the following to the 3.4 section.

*Line348: "Different models use different algorithms so the important factors may differ slightly. We have also tried using Gradient Boosted Decision Trees (GBDT) (Friedman, 2001), and their top three important factors are "Aspectcosine", "MSWEP" (Precipitation), and "Downward shortwave radiation". So this model choice does not affect our conclusions."*

Actually, "sklearn" and "scikit-learn" are the same package ([https://en.wikipedia.org/wiki/Scikit-learn](https://en.wikipedia.org/wiki/Scikit-learn) ). For consistency, we replaced "sklearn" with "scikit-learn".

*The author mentioned the driest sites are hard to predict, and suggested the reason as scarce but sudden rainfall events. Do authors believe it **may be related to the loss function** used in the LSTM model (i.e., some loss functions would not emphasize the extreme high/low values)? I also don't follow the logic*

*behind Line 345. From the error type analysis, the authors mentioned the comparison between temporal and spatial tests, especially for error type B (nonstationary). **But the boxplots of R** from the temporal test and spatial test over driest sites are similar to me, with a median R of approximately 0.7.*

Regarding the loss function, we are inclined to think this is not the case, as our experience has been that the tradeoff due to loss function tends to be small, that is, the training would typically reduce all kinds of error. Because LSTM does not have strong functional form and does not respect mass balance, the structural tradeoff is mild.

Regarding the part about line 345, we would like to clarify that *Figure 6-I-h* and *Figure 6-II-h* are the same figures plotted using temporal test error or spatial test error as the variable plotted (not as "target" as originally explained in the caption, as this figure does not involve a model). Here we showed two types of error just to show that our conclusion is more or less robust, with some nuanced differences between these two error types. For panel h, the readers should focus on the trend going from dry to wet, where the temporal test metric shows a clear rising trend (so dry sites were poor) while the spatial test does not have such a strong trend, as shown below.

*Below is the original text with track changes highlighting the changes we will made:*
*The model correlation in the temporal test generally rises as soil moisture goes up, until reaching the wettest regime (0.48-0.6), where its variability increases (Figure 6-I-h). The sites in the middle range tend to have continuity in soil moisture and regular rainfall patterns, which are most ideal for LSTM. There is a clear rising trend for R of temporal test from dry to wet sites. The driest sites may be difficult to predict due to scarce but sudden rainfall events that quickly dry out, which reduces the usefulness of LSTM's memory capability. When we plotted the spatial test R (Figure 6-II-h), the pattern is similar but less pronounced, which suggests the driest sites are also more impacted by temporal non-stationarity than spatial heterogeneity, because they have seen limited storm events. Toward the wettest regime, saturation often occurs, and soil moisture may be influenced by groundwater processes which are difficult to account for.*

[Figure]

Figure 6-I-h, we see a clear rising trend of temporal test R from dry to wet sites.

[Figure]

Figure 6-II-h, the trend is less noticeable.

*Line 146 mentioned the use of different resolutions for the static terrain attributes. What is the aspect resolution used in the random forest model?*

Actually, "… from the Global 1,5,10,100-km Topography database" means that the website provides data in different resolutions. However, we only downloaded 10-km data and used the bilinear method to get 9-km resolution input data, so our elevation and aspect only have the 9-km resolution. We will add the following to the paper:

*Line 147: "We changed their resolution to 9-km using the bilinear interpolation method."*

*The code files in the Zenodo repository are not enough to replicate the experiments. I'd suggest the authors update their repository, either during or after the peer-review process.*

We have updated the code in Zenodo. After testing by other folks, the code works perfectly with or without GPU. Since the global input data is 900 GB, it is not practical to upload all data to Zenodo.

---

## Author Response (AR1)

We thank the editor for handling the manuscript, and the reviewers for their valuable comments. In the last round, no major revision to the computational results were requested. Most of the comments are regarding clarifications and writing. We did run a boosted regression tree instead of random forest model to verify that the factorial results are robust. We have revised the manuscript accordingly. The line numbers refer to the lines in the revised manuscript.

**Reviewer1:**

*The manuscript "Evaluating a Global Soil Moisture dataset from a Multitask Model (GSM3 v1.0) for current and emerging threats to crops" presents a LSTM-based machine learning model for global soil moisture estimation. The authors used a multi-loss framework to train the LSTM model, and employed multiple reference datasets for evaluation. Specifically, the work investigated spatial generalization ability by cross-validations, both randomly and continent-based sampling. The overall quality of the work is excellent and fits in the scope of GMD. With that, I do have the following comments.* Thank you for your evaluation.

*The title is somewhat confusing to me. It indicates the authors also care about crops in addition to soil moisture, as soil moisture will affect the crops. So I was expecting to read something about agricultural applications (e.g., use the DL-based soil moisture to drive some crop models). However in the manuscript, it is only mentioned before the conclusion section that this model will be put into production, and the major focus of the paper is model benchmarks on soil moisture. I suggest the authors have more discussion on crop applications, or revise the title to make it clearer. In addition, if the main focus is to evaluate the dataset instead of the LSTM model, I suggest making the dataset publicly available and adding a link to access this dataset.*

We understand the title may be a little confusing. The title now reads
"Evaluating a Global Soil Moisture dataset from a Multitask Model (GSM3 v1.0) with potential application to crop threats"

The overall motivation of this work, and the way it will be used, is to support the detection and prediction of threats (like pests) to crops in a region like Africa so we hope the mission is reflected in the title. However, to more clearly define the scope of this paper, we decided to follow Reviewer 2's suggestion to add "with potential application". The code and dataset were uploaded to the online archive Zenodo at the time of initial submission, and a link to this archive is contained in the "Code/Data Availability section". https://doi.org/10.5281/zenodo.7344484

*To clarify, what's the difference between the SoMo.ml model and the authors' LSTM model? Do they share the same dynamic and static input variables with the only difference as the loss function? In Figure 1 and 2, the authors compared the metrics derived from the training period (Multitask_train and SoMo.ml_train). The R value or RMSE from the whole training period is not important, as one can always overfit the model. I understand that a barplot for Multitask_train shows the model is not overfitting, but a comparison between Multitask_temporal and SoMo.ml_temporal would make more*

*sense to me.*

There are three major differences between SoMo.ml and multitask models:

1. SoMo.ml used 13 different types of dynamic and static data as input data, while multitask used 35 different data types, and their data sources and resolutions are also different.
2. More importantly, the model's loss calculation is different. SoMo.ml was trained using ~1000 sites worldwide, so the loss is calculated based on the in-situ data. The multitask model was trained with both in-situ and satellite grid data, and the loss is calculated based on both in-situ and satellite data.
3. The model structure is different too. SoMo.ml uses the input data from day t-364 to day t to obtain the target soil moisture data on day t, which is a sequence-to-one structure. The multitask model uses the input data from day t-364 to day t to obtain the target soil moisture data from day t-364 to day t. It is a sequence-to-sequence structure.

We have added the following to the paper:

*Line: 186: "The SoMo.ml model also differs from the multitask model as it uses different input data, only in-situ data in calculating the loss function, and a sequence-to-one structure. Despite these differences, we still think a best effort at comparison could be useful to the community."*

In addition, SoMo.ml only provided the final products which used all data. We cannot access their temporal test results and thus cannot make that comparison. More importantly, their model was retrained using all available data, so they show the result of the training period while we show results from the testing period. Even in this unfavorable situation, we still achieved a good performance. This question has been answered in the paper:

*Line: 183: "Notably, the SoMo.ml product provides soil moisture estimation from 0-10 cm depth, not 0-5cm depth. Its final product was obtained by retraining the model using all available sites and times rather than by using spatial cross-validation (spatial cross-validation is regarded as a more rigorous test, so this comparison puts our model at a disadvantage)"*

*What is ubRMSE in Table1? Does Corr represent Pearsonr correlation coefficient? In Figure 3 and 4, the meanings of colormaps are not the same. In the correlation map, greener represents better performance, but it is worse performance in the RMSE plot. I would suggest a uniform colorbar with light (dark) colors for high performance and dark (light) colors for low performance.*

ubRMSE is unbiased RMSE. We have added section 2.6, Evaluation Metrics to explain the meaning of each metric.

*Line 223: "The metrics used to evaluate the Multitask model's performance include Pearson's correlation coefficient (Corr), bias, root-mean-square error (RMSE), and unbiased RMSE (ubRMSE), in which RMSE is calculated after bias is removed. These metrics are the median value of all satellite grids and in-situ. When we calculate these metrics, we remove the observed and predicted data when there is a nan value (not a number; an error) in the observation."*

We have changed the colorbars in Figure 3, Figure 4, and Figure S1 to be uniform, with dark colors for high performance and light colors for low performance.

*The LSTM model is optimized towards both the in-situ estimation and the satellite products. In Figure1 when comparing the model performance, the authors selected the SMAP and GLDAS products, which are gridded datasets. When evaluated against the ISMN dataset, it is not a fair comparison because of the coarse resolution of SMAP. How do the authors correlate in-situ measurements with gridded datasets? Will the result change when switching to 25-km resolution (i.e., with coarser resolution, the dataset will lose more representations)? A further question is that, what is the meaning of the LSTM outputs? Is it the best estimation over the grid points (e.g., an average over the grid spacing), or the best estimation for the in-situ observations?*

We were not really attempting to outcompete SMAP which, as noted, are coarsely gridded observations and are used as training data. It would be impossible to compare to SMAP or GLDAS optimized for comparison with sparse in-situ data because such products do not exist. However, we think the community will benefit from having this comparison in the Figure to serve as a helpful context to know where our model stands. To clarify this, we have added the following:

*Line 178:* "*It is to be noted that SMAP and GLDAS products were not optimized to match the sparse in-situ networks so this comparison is not entirely fair, but they were shown to provide context.*"

The meaning of the LSTM product is the average soil moisture for the 9-km grid. The multitask model input data include satellite grid data (9-km) and in-situ data, where the in-situ forcing, and static attribute data are extracted for the 9-km satellite grid. The model's prediction resolution is 9-km, so it is fair to directly compare the performance of SMAP and the model at the ISMN location. However, the GLDAS's resolution is 0.25 degrees, Corr is 0.609, Bias is 0.045, RMSE is 0.101, and ubRMSE is 0.069. For a fair comparison, we adjusted the multitask model's resolution to 0.25 degrees and resampled the other products to 0.25 degrees (Table S5). Therefore, the model's performance gets slightly worse as the resolution gets coarser, but it does not change our conclusions.

We have added the following to the paper:

*Line 191:* "*We also resampled the model's input data and the other products to retrain a new model. They were compared at the same resolution of 0.25 degrees. The model's performance dropped slightly but the results supported the same conclusions as the 9-km resolution (Table S5).*"

We trained the multitask model using the satellite grid and in-situ data. The model is still an LSTM model, but its loss values are calculated from the satellite loss and in-situ loss. Therefore, the LSTM model's output depends on its inputs when predicting soil moisture. If the input data is satellite grid data, then the output is also grid soil moisture prediction. If the input data is in-situ, the output is in-situ location soil moisture prediction. In this paper, we used in-situ inputs to obtain the soil moisture predictions to compare with ISMN. We used global grid data for the final products to get the global 9-km soil moisture from 2015 to 2020. We have added the following to the paper:

*Line 188: "The model performance under different experiments is compared with the ISMN in-situ data, while the final product input and output data are both global 9-km grid data."*

*The authors split the global dataset into 7 continents. Would it be a more straightforward comparison to have a similar 7-fold cross validation to match the number of continents in Section 3.2? I believe 5-fold is a common choice but 7-fold may be a more intuitive and fair comparison.*

In the paper, we used a 5-fold method for the global data to get the results: Corr is 0.792, Bias is -0.0003, RMSE is 0.075, and ubRMSE is 0.056. When we trained the model using 7-fold, Corr is 0.797, Bias is -0.0004, RMSE is 0.075, and ubRMSE is 0.056. The results did not differ significantly. We have added the following to the paper:

*Line 197: "We also ran a 7-fold cross validation experiment. However, there was no significant difference in their results. To save computational resources, we showed the results from the 5-fold experiments."*

*When analyzing the factor controls, the authors selected a random forest model. What is the performance/skill of this random forest model? Is Figure 5 using spatial R as target or temporal R, or is it a multi-output model? At line 329 it shows the target is temporal R, but line 330 shows R is from either temporal or spatial tests.*

We have added the following to the paper:

*Line 210: "A Random Forest (RF) model is a classification/regression algorithm consisting of many decision trees that use bagging and randomness of features to create a series of decision trees. It is suitable for non-linear data and reduces the risk of over-fitting."*

*Line 352: "The RF model has a test correlation of 0.6 (with 80% training data and 20% test data) but its only purpose here is to provide a reading on the top three factors."*

*Line 155: "To perform factorial importance analysis, we also calculated long-term averages of daily LST, Albedo, LST, and SMAP data and used them along with other static attributes as inputs in the LSTM model and Random Forest model (using R of the tests as the target, removed duplication)"*

For Figure 5, we only used temporal experiment R, so we modified the caption of Figure 5 to *"The feature importance determined from a Random Forest (RF) model constructed to predict temporal test R using all the unduplicated category data presented in this paper as inputs"*.

In the main text, we also modified *"... and R from either temporal or spatial tests as the targets."* to *"... and R from the temporal test serves as the target."* (Line 357)

*Regarding the random forest model, is the conclusion independent of the model choice (or what's the reason to choose a random forest model)? Will we get different results if we switch to extreme boosted trees or other tree-based models? Also please check the name of the python package. It is used as "sklearn" in python but the paper referenced (Pedregosa et al., 2011) shows "scikit-learn" in the title.*

Different models may give somewhat different but qualitatively the same results. We built Gradient Boosted Decision Trees (GBDT) to analyze the important factors. "Aspectcosine", "MSWEP" (precipitation), and "Downward shortwave radiation" are the top three ranking factors in the results. Compared with Random Forest, precipitation ranked higher than SMAP in Boosted Trees, but this does not affect our conclusions. Because the precipitation and soil moisture trends are the same, they are equally represented. We have added the following to section 3.4.

*Line 353: "We have also tried using Gradient Boosted Decision Trees (Friedman, 2001), which produced a test correlation of 0.77, and the top three important factors were slope aspect, precipitation, and surface solar radiation downwards."*

Actually, "sklearn" and "scikit-learn" are the same package (https://en.wikipedia.org/wiki/Scikit-learn ). For consistency, we replaced "sklearn" with "scikit-learn".

*The author mentioned the driest sites are hard to predict, and suggested the reason as scarce but sudden rainfall events. Do authors believe it **may be related to the loss function** used in the LSTM model (i.e., some loss functions would not emphasize the extreme high/low values)? I also don't follow the logic behind Line 345. From the error type analysis, the authors mentioned the comparison between temporal and spatial tests, especially for error type B (nonstationary). **But the boxplots of R** from the temporal test and spatial test over driest sites are similar to me, with a median R of approximately 0.7.*

Regarding the loss function, we are inclined to think this is not the case, as our experience has been that the tradeoff due to loss function tends to be small, that is, the training would typically reduce all kinds of error to the extent possible. Because LSTM does not have strong functional form and does not respect mass balances, the structural tradeoff is mild. We expect to see more tradeoff for process-based models which have structural constraints.

Regarding the part about line 345, we would like to clarify that *Figure 6-I-h* and *Figure 6-II-h* are the same figures plotted using temporal test error or spatial test error as the variable plotted (not as "target" as originally explained in the caption, as this figure does not involve a model). Here we showed two types of error just to show that our conclusion is reasonably robust, with some nuanced differences between these two error types. For panel *h*, the readers should focus on the trend going from dry to wet, where the temporal test metric shows a clear rising trend (so dry sites were poor) while the spatial test does not have such a strong trend, as shown below.

We have modified the original text as shown below, with changes in bold font:
*The model correlation in the temporal test generally rises as soil moisture goes up, until reaching the wettest regime (0.48-0.6), where its variability increases (Figure 6-I-h). The sites in the middle range tend to have continuity in soil moisture and regular rainfall patterns, which are most ideal for LSTM. **There is a clear rising trend in R for the temporal test, from dry to wet sites.** The driest sites may be difficult to predict due to scarce but sudden rainfall events that quickly dry out, which reduces the usefulness of LSTM's memory capability. **When we plotted** the spatial test R (Figure 6-II-h), the pattern is similar but less pronounced, which suggests the driest sites are also more impacted by temporal non-stationarity than spatial heterogeneity, because they have seen limited storm events. Toward the wettest regime, saturation often occurs, and soil moisture may be influenced by*

*groundwater processes which are difficult to account for.*

[Figure]

Figure 6-I-h, we see a rising trend of temporal test R from dry to wet sites (left to right).

[Figure]

Figure 6-II-h, the trend is less noticeable.

*Line 146 mentioned the use of different resolutions for the static terrain attributes. What is the aspect resolution used in the random forest model?*

Actually, "… from the Global 1,5,10,100-km Topography database" means that the website provides data in different resolutions. However, we only downloaded 10-km data and used the bilinear method to get 9-km resolution input data, so our elevation and aspect only have the 9-km resolution. We have added the following to the paper:

*Line 147: ", and we changed their resolution from 10-km to 9-km using the bilinear interpolation method."*

*The code files in the Zenodo repository are not enough to replicate the experiments. I'd suggest the authors update their repository, either during or after the peer-review process.*

We have updated the code in Zenodo. After testing by others, the code works perfectly with or without GPU. Since the global input data is 900 GB, it is not practical to upload all data to Zenodo.

**Reviewer2:**

This paper presents a long short-term memory (LSTM) recurrent neural network model for use in global predictions of soil moisture. The model learns from both satellite-based remote sensing and in-situ soil moisture data sets, adopting a multi-task learning approach that optimizes a linear combination individual loss functions that are computed using satellite products (SMAP L3 at 9 km nominal resolution) and in-situ data (from the International Soil Moisture Network). The authors present systematic comparisons of the model with the SMAP product, process-based land surface models, and another LSTM model (SoMo.ml) that is trained exclusively with in-situ observation network data to demonstrate the utility and limitations of the model.

The subject of this paper is a good fit for GMD, and the paper's quality is excellent, with very good reproducibility, model evaluation, discussion, and writing. I have little to suggest that could improve the paper without expanding its scope (which would be unfair to the authors and is not necessary). I have only two minor criticisms / comments to mention:

Thank you for your evaluation.

I believe that the title is misleading because of the phrase "for current and emerging threats to crops". I understand that one of the main motivations for developing this model is for it to serve as part of what might be termed an early-warning system for threats to crops in Africa. However, this system does not yet appear to be fully in production, or, if it is, the authors have not given a full description or examples of its application. An accurate soil moisture model has clear benefits (and possibly is an unavoidable necessity) for such a system. But since there is no substantial discussion of how the model is being used in this context, I suggest either dropping the phrase "for current and emerging threats to crops" or modifying the title to change the "for" to "with potential application to". (I see that Anonymous Referee #1 has raised the same concern.)

We agree that the title as it was could be seen as misleading. We appreciate your suggestion, and have implemented it. The title now reads "Evaluating a Global Soil Moisture dataset from a Multitask Model (GSM3 v1.0) with potential application to crop threats."

I was confused by "Normalized Difference Vegetation Index" (NDVI) being listed as one of the static landscape attributes in the paragraph beginning at line 147. I have worked extensively with NVDI time series (in the context of emerging threat detection, incidentally) and it is something that I think of as the antithesis of a static landscape attribute (outside of a place like a barren desert environment). I had to look at the author's code in their Zenodo repository to see that this is actually NVDI averaged over some long time period (though I didn't dig enough to figure exactly what this time period is or how the average is computed). I would appreciate it if the authors would amend the paper to add something about what sort of NVDI average is being used here.

We have added the following to the paper:

*Line 153: "We averaged all NDVI data from April 1, 2015 to March 31, 2022 to obtain multiple years of static NDVI data, and resampled the data to 9 km using bilinear interpolation."*